# The expression of essential selenoproteins during development requires SECIS-binding protein 2–like

Nora T Kiledjian, Rushvi Shah ⓘ, Michael B Vetick, Paul R Copeland ⓘ

**The dietary requirement for selenium is based on its incorporation into selenoproteins, which contain the amino acid selenocysteine (Sec). The Sec insertion sequence (SECIS) is an RNA structure found in the 3′ UTR of all selenoprotein mRNAs, and it is required to convert in-frame UGA codons from termination to Sec-incorporating codons. SECIS-binding protein 2 (Sbp2) is required for Sec incorporation, but its paralogue, SECIS-binding protein 2–like (Secisbp2l), while conserved, has no known function. Here we determined the relative roles of Sbp2 and Secisbp2l by introducing CRISPR mutations in both genes in zebrafish. By monitoring selenoprotein synthesis with $^{75}$Se labeling during embryogenesis, we found that $sbp2^{-/-}$ embryos still make a select subset of selenoproteins but $secisbp2l^{-/-}$ embryos retain the full complement. Abrogation of both genes completely prevents selenoprotein synthesis and juveniles die at 14 days post fertilization. Embryos lacking Sbp2 are sensitive to oxidative stress and express the stress marker Vtg1. We propose a model where Secisbp2l is required to promote essential selenoprotein synthesis when Sbp2 activity is compromised.**

## Introduction

Selenium is an essential trace element that is incorporated as the amino acid selenocysteine (Sec) into ~20–50 vertebrate proteins known collectively as selenoproteins. This class of proteins serves a variety of disparate functions, for example, resolving oxidative stress (the glutathione peroxidases) and hormone synthesis (iodothyronine deiodinases). Because Sec is encoded by in-frame UGA codons that would otherwise be interpreted as termination codons, a dedicated set of signals and factors are required to allow Sec incorporation (reviewed in Howard and Copeland [2019]). In every selenoprotein mRNA, there is a stem-loop feature in the 3′ UTR, the Sec insertion sequence (SECIS), that autonomously converts upstream in-frame UGA codons to specify Sec (Berry et al, 1991). In addition, a SECIS-binding protein (Sbp2) binds specifically to the SECIS element and recruits the ternary complex consisting of

the Sec-specific translation elongation factor (Eefsec), Sec-tRNA$^{Sec}$, and GTP (Copeland et al, 2000; Tujebajeva et al, 2000; Donovan et al, 2008). These factors are known to be sufficient to allow Sec incorporation in a plant in vitro translation system, which is otherwise devoid of selenoprotein related factors (Gupta et al, 2013).

Although Sbp2 is known to be necessary and sufficient for Sec incorporation in vitro, all vertebrates possess a paralogous gene called *secisbp2l*, which also binds to all SECIS elements (Donovan & Copeland, 2012). Sbp2 and Secisbp2l share similar domain structure, consisting of an N-terminal domain with no known function, a central domain required for Sec incorporation that works together with the downstream RNA-binding domain required for SECIS binding. In addition, Secisbp2l contains a C-terminal glutamate rich domain that is not present in Sbp2. Despite the overall similarity between the two, Secisbp2l does not support Sec incorporation in vitro (Donovan & Copeland, 2012), so its function remains unknown. Interestingly, because Secisbp2l is preferentially expressed in epithelial cells (Kapushesky et al, 2010) it might be expected to play a role in responding to external stimuli. In fact, Secisbp2l expression is strongly positively correlated with protection from lung adenocarcinoma, which underscores the importance of determining function (McKay et al, 2017).

In the context of vertebrate development, elimination of SBP2 in mice resulted in lethality during gastrulation (Seeher et al, 2014), and subsequent studies showed that this block in development could be overcome by expressing a single selenoprotein (GPX4) with a Cys residue substituted for Sec (Ingold et al, 2018). In addition, the muscle-specific selenoprotein Selenon was reported to be required for normal muscle function and calcium flux in zebrafish embryos (Deniziak et al, 2007; Jurynec et al, 2008), and in situ hybridization revealed a complex array of tissue specific expression for 21 selenoprotein mRNAs during zebrafish development (Thisse et al, 2003). Despite these efforts, very little is known about the regulation of selenoprotein synthesis during development, so here we have used the zebrafish model system to determine the relative roles of Secisbp2l and Sbp2, thereby establishing a highly tractable system in which to study the mechanism of Sec incorporation and the consequences of selenoprotein deficiency in vivo. The largest selenoproteomes exist in bony fishes, and zebrafish has been reported to possess 38 selenoprotein genes, albeit with

Rutgers–Robert Wood Johnson Medical School, Piscataway, NJ, USA

Correspondence: paul.copeland@rutgers.edu

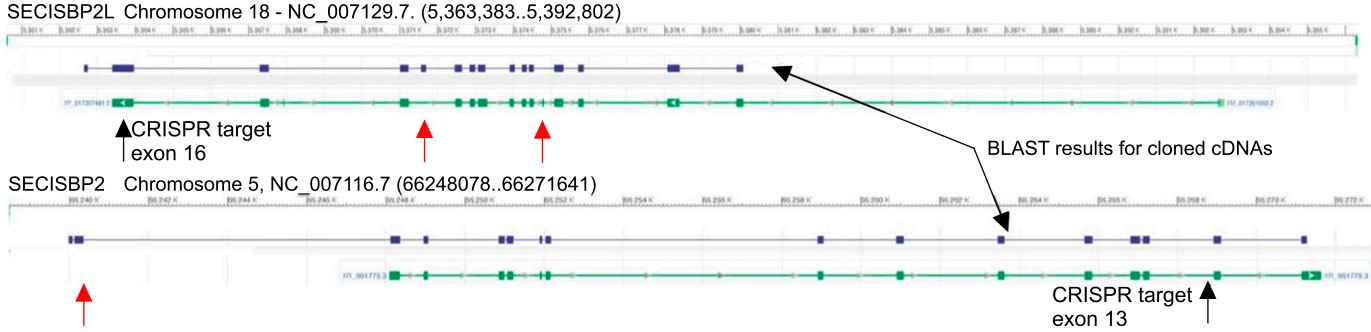

**Figure 1. Diagram of the zebrafish *secisbp2l* and *sbp2* gene loci (NCBI; note reverse orientation of *secisbp2l*).**
The results of BLAST analysis using the cloned cDNAs for both genes is shown in purple with differences between annotated exons (green) and those derived from cDNA cloning marked by red arrows. The CRISRP/Cas9 target sites in the RNA-binding domains are indicated with arrows.

several duplications (Mariotti et al, 2012). Considering the accessibility of developmental biology in the zebrafish system and the relative paucity of literature about the role of selenoproteins during vertebrate development, a lot can be learned by leveraging the tractability of this system to answer key questions surrounding selenoprotein synthesis and function.

In this report, we have used CRISPR/Cas9 methodology to generate zebrafish lacking either Sbp2 or Secisbp2l. Although no overt phenotypes were observed, we found that most but not all selenoprotein production was substantially reduced in the *sbp2*$^{-/-}$ larvae, and they were significantly more sensitive to oxidative stress. *Secisbp2l*$^{-/-}$ animals showed slightly reduced selenoprotein production, which was completely inhibited and led to death at 14 d post fertilization (dpf) when both genes were ablated.

## Results

### *sbp2* and *secisbp2l* genes

Unlike many genes in zebrafish, those encoding Sbp2 and Secisbp2l are present as single copy genes. The RefSeq RNA entry for zebrafish *secisbp2l* has a 2,721 nt coding region and is missing several segments relative to other fish species such as goldfish (*Carassius auratus*), which has a coding region of 3,012 nt. Using total RNA purified from zebrafish embryos grown to 5 dpf, we attempted to clone the cDNA corresponding to zebrafish *secisbp2l*, but we were not able to obtain any single products corresponding to the full length sequence. However, we were able to clone overlapping fragments to obtain a candidate full length sequence with a 3,085 nt coding region that is 85% identical to the goldfish sequence (see the Materials and Methods section). The case of *sbp2* is similar where the RefSeq RNA entry lacked the N-terminal portion of the mRNA relative to other species (Fig 1). The only annotated sequence that contains a predicted N-terminal region with sequence upstream of the conserved C-terminal domains was also found in goldfish. Interestingly, the N-terminal ~80 amino acids of the predicted goldfish Sbp2 are only found in the Clupeocephala super cohort of teleost fish, which includes the cyprinidae (zebrafish and goldfish). The large (~1.5 kb) N-terminal sequence that is found in both Sbp2 and Secisbp2l in most species (Donovan

& Copeland, 2009](#)) is only present in Clupeocephala Secisbp2l but not in Sbp2. Using total RNA purified from larvae grown to 5 d and primers corresponding to the regions annotated as the 5′ and 3′ regions of the goldfish sequence, we cloned the cDNA corresponding to zebrafish *sbp2,* which yielded a 2,274-nt coding region that is 83% identical to the goldfish sequence, thus confirming the presence of the short and unique N-terminal domain in zebrafish sbp2. Fig 1 shows a diagram of alignments between the cDNA clones we obtained at 5 dpf and the NCBI gene sequences, highlighting discrepancies between the gene annotation and the sequences obtained by RT-PCR. Note that the official name for sbp2 is secisbp2, but in this report we will use "sbp2" so it is easily distinguished from "secisbp2l" in the text.

### Selenoprotein mRNA expression during zebrafish development

To assess the expression of selenoprotein mRNAs and those that are required for selenoprotein synthesis during zebrafish development, we mined existing RNA-seq data obtained by the Busch-Nentwich group at Wellcome Sanger Institute (White et al, 2017). Fig 2 shows a heat map of selenoprotein mRNA expression as well as Sec incorporation factor mRNA expression across a non-linear range of early developmental time covering 0–120 hours post fertilization (hpf). Sec incorporation factor mRNA expression is generally low, particularly for *eefsec* (Sec-specific translation elongation factor) and *pstk* (the enzyme that phosphorylates the Ser intermediate in Sec synthesis). It is notable that both *sbp2* and *secisbp2l* are expressed at low levels across the entire time range with small peaks of expression at 2–4 hpf and again at 96 hpf, and these peaks are generally correlated with higher levels of selenoprotein mRNAs. In addition to the selenoprotein mRNAs that have very low expression across the entire range, there are two other classes of selenoprotein mRNA expression: (1) early high expression (0–10 hpf above 200 transcripts per million), and (2) late high expressing (24–120 hpf above 200 transcripts per million). It is interesting to note that in cases where the increase in selenoprotein mRNA expression is very high (e.g., *selenop*), there is no correspondingly significant increase in mRNAs encoding *sbp2* and *eefsec*, which are directly required for inserting Sec during translation. This dataset establishes that selenoprotein mRNA expression is biased toward larval development from 1 to 5 dpf, most

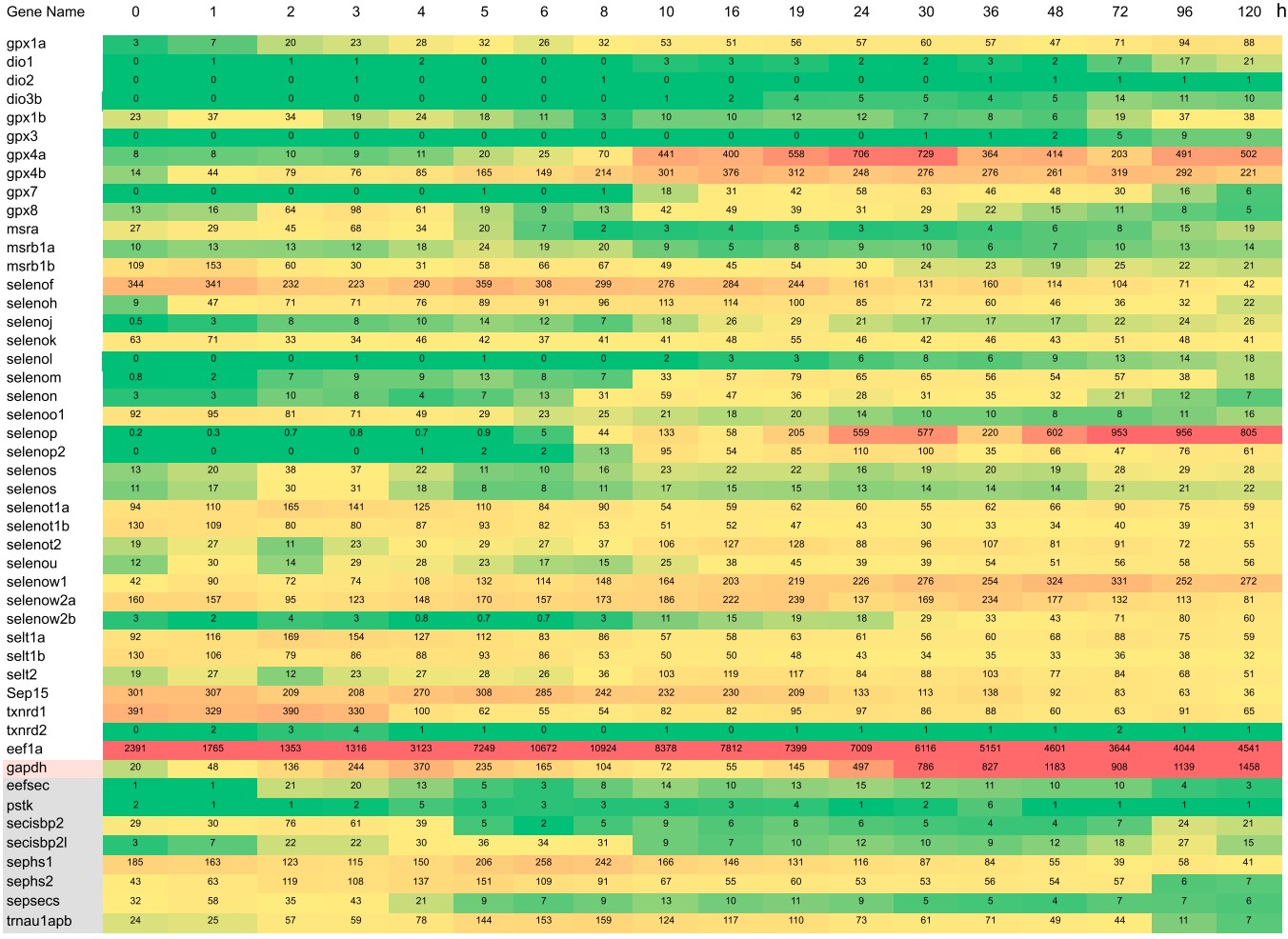

**Figure 2. Sec-related gene expression during zebrafish development.**
Heat map of existing transcriptomic data (White et al, 2017) highlighting selenoprotein and selenoprotein synthetic factor mRNA expression during zebrafish development. Numbers correspond to average transcripts per million from four samples. The color coding is a linear gradient from green to red where the 50th percentile is yellow with the maximum set at 805, which is the maximum for selenoprotein mRNA expression.

notably through expression of *selenop* and *gpx*4. These data also highlight the fact that selenoprotein mRNA transcription and/or stability is highly regulated during this time period, highlighting the significance of temporal regulation that cannot be studied in vitro or in cells.

## Gene disruption and genomic analysis

We sought to determine the relative contributions of Sbp2 and Secisbp2l to selenoprotein production in vivo by ablating their expression in zebrafish. For *secisbp2l*, a single guide RNA (sgRNA) was designed to target the RNA-binding domain at position 2,325–2,344 (encoding KLVSLT, which is KLVELT in human; Fig 1). For *sbp2*, we used separate tracr and crispr RNAs, the latter also targeting a conserved region in the RNA-binding domain within exon 13 at position 1,955–1,977 (encoding VPVSL which is VPVLS in human). The 3′ biased locations were selected to avoid the possibility of downstream translation initiation that may bypass indels in the N-terminal half of the genes. This is necessary because the

C-terminal halves of Sbp2 and Secisbp2l are functional (Donovan & Copeland, 2012). Injected embryos were screened by genomic PCR and sequencing of the targeted area revealed two edited alleles for *secisbp2l* with 19 and 26 base pair (bp) insertions, both of which caused a frameshift and introduction of a premature termination codon at the position of insertion. We chose the 26-bp insertion allele for further analysis. In the case of *sbp2*, we obtained both 1 and 5 bp deletions. We chose the 5 bp deletion allele for further analysis. Outcrossing F0 founders to wt fish generated F1 heterozygotes that were used for inbreeding to create stable homozygous lines for the 26-bp insertion for *secisbp2l* and 5-bp deletion for *sbp2*. Edited alleles were confirmed through sequence analysis of genomic DNA (Fig 3). In addition, sequence analysis of RT-PCR products did not reveal any traces of wild-type sequence for either *sbp2* or *secisbp2l*. No discernible overt phenotypes were observed for either *sbp2*[−/−] or *secisbp2l*[−/−]. This was an unexpected result in the case of Sbp2 because mouse embryos that lack Sbp2 expression die before gastrulation (Seeher et al, 2014). Note that the *sbp2*[−/−] or *secisbp2l*[−/−] strains were generated on different genetic

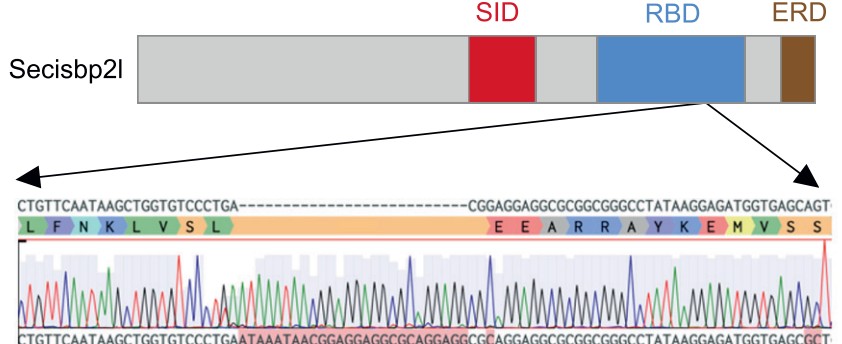

**Figure 3. CRISPR/Cas9 mutagenesis of the *secisbp2l* and *sbp2* loci.**
Domain diagrams illustrating the relative positions of the Sec incorporation domain (SID), RNA-binding domain (RBD) and the Glu-rich domain (ERD), and sequence analysis of *secisbp2l* and *sbp2* mutations generated by CRISPR/Cas9 illustrating the 26-bp insertion for *secisbp2l* and the 5 bp deletion in *sbp2*.

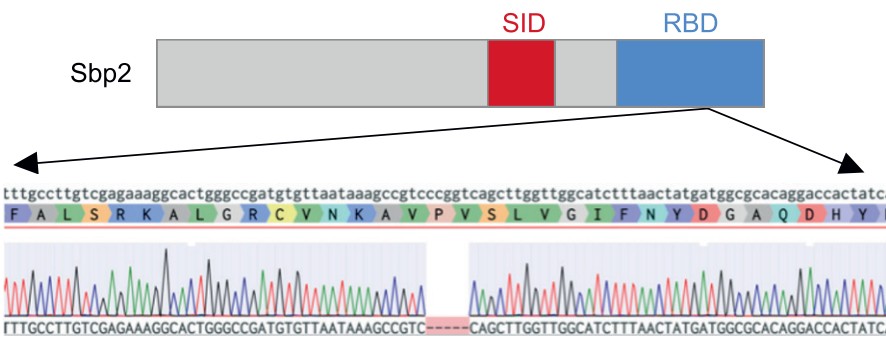

backgrounds so all comparisons described in this report are between the mutant line and its isogenic wild-type sibling line.

### Analysis of disrupted Sbp2 expression by immunoblot

Having established that the *sbp2* and *secisbp2l* genes were disrupted, we sought to examine protein expression across development in the wild-type and mutant lines. For Sbp2 analysis, we were able to use an affinity purified commercial antibody raised against a C-terminal portion of the human protein (aa 506–854), which is 79% identical in the conserved RNA-binding domain. Fig 4 shows immunoblot analyses across 3 d of development. Whereas the predicted size of full-length zebrafish Sbp2 is only 85 kD, we observed a high-intensity band migrating at ~130 kD that is not present in the *sbp2*$^{-/-}$ strain. Aberrant migration was expected because mammalian Sbp2 has a predicted molecular weight of 95 kD but endogenous and recombinant Sbp2 migrates at ~120 kD in SDS–PAGE (Copeland et al, 2000). At 5 dpf, the *sbp2* signal in wt larvae is substantially reduced relative to the tubulin control. This was not expected because the transcriptomic analysis indicated that the peak of *sbp2* mRNA expression occurred at days 4 and 5 (see Fig 2). Note the presence of a ~120 kD band that appears only in the lanes containing sbp2$^{-/-}$ lysate. It is possible that this represents the truncated protein resulting from the 5-bp deletion that would yield a 12-kD truncation at the C-terminal end. Extensive mutagenesis of mammalian Sbp2 has established that this highly conserved C-terminal sequence that is missing in the mutant line is essential for Sbp2 function (Copeland et al, 2001; Caban et al, 2007). For Secisbp2l analysis we raised a polyclonal

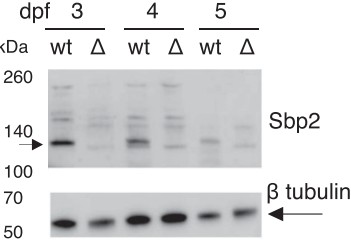

**Figure 4. Immunoblot analysis of Sbp2 expression.**
Wild-type (wt) and *sbp2*$^{-/-}$ (Δ) embryos at the indicated days post fertilization were lysed and two embryo equivalents were loaded onto a 4–12% gradient SDS–PAGE gel. Immunoblot was probed with a polyclonal antibody raised against human SBP2.

antibody against a C-terminal fragment of the predicted zebrafish protein (aa 462–811) but we were not able to detect a candidate protein.

### Selenoprotein expression is differentially reduced in *sbp2*$^{-/-}$ and *secisbp2l*$^{-/-}$ embryos

To assess selenoprotein expression during early zebrafish development, we used $^{75}$Se-selenite metabolic labeling. Because mRNA expression data show substantial selenoprotein mRNA expression from days 3 to 5, we focused on this time frame. To analyze selenoprotein expression in embryos that lack Sbp2, we screened 5 dpf larvae that resulted from a cross between heterozygous *sbp2*$^{+/-}$ animals so that we might observe the phenotypes of the three

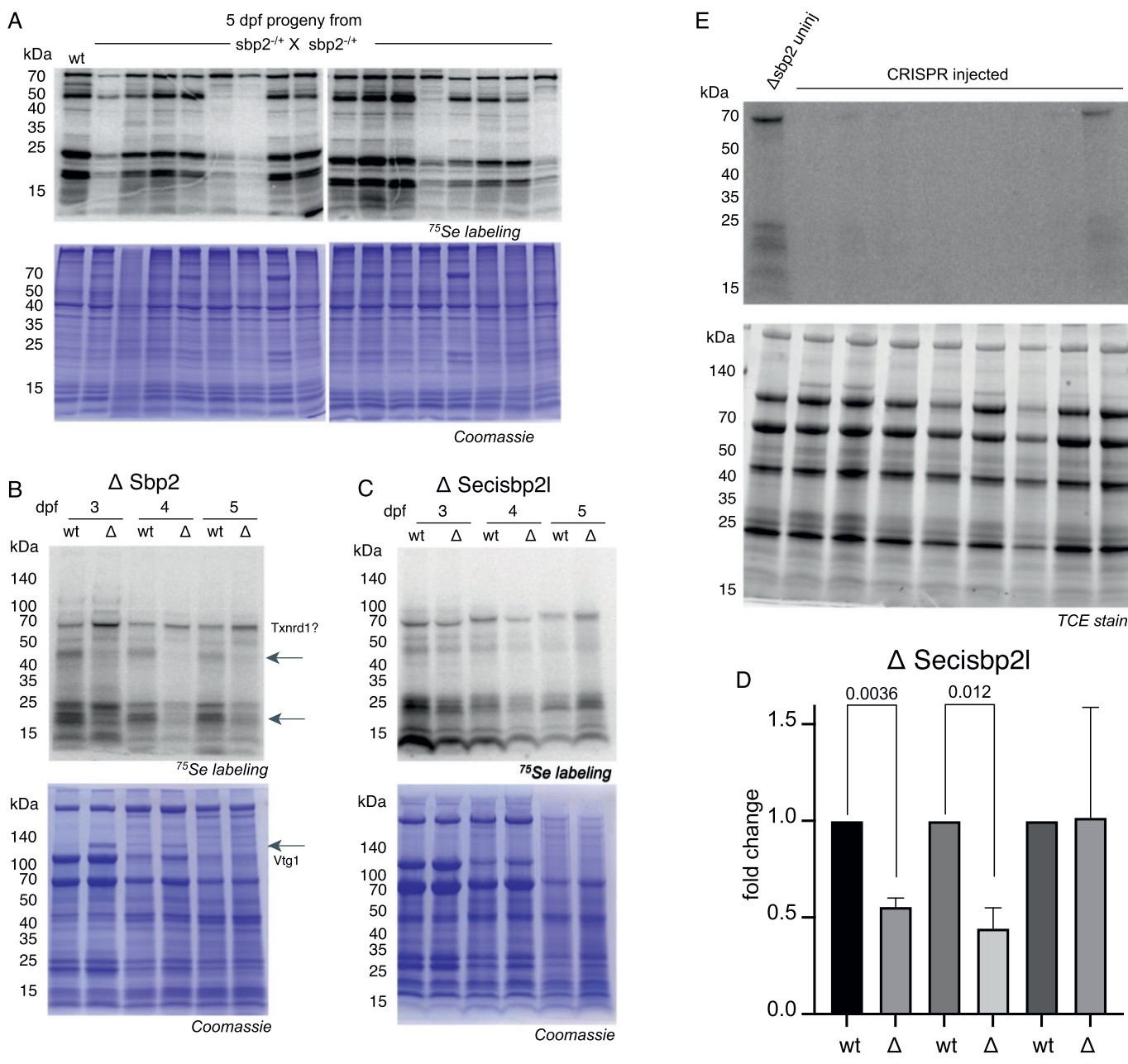

**Figure 5. Metabolic labeling reveals selenoprotein synthesis defects.**
**(A)** $sbp2^{+/-}$ heterozygous fish were mated and offspring were incubated with 375 nM $^{75}$Se for 24 h at 4 days post fertilization (dpf). 16 larvae were randomly chosen for SDS–PAGE and phosphorimaging analysis (top panels) and the same gels were stained with Coomassie blue (bottom panels). **(A, B)** $sbp2^{-/-}$ and $secisbp2l^{-/-}$ larvae at the developmental stage indicated were labeled and analyzed as described in (A). The 68-kD band that is not affected by the loss of Sbp2 is predicted to be Txnrd1 based on molecular weight. The arrows point out the bands with marked decrease in intensity in the mutated line. The increase in Vtg1 in 3–4 dpf larvae is noted in the Coomassie-stained gel (bottom panel). **(B, C)** Quantitation of the ΔSecisbp2L gel in (B). Three replicate gels were analyzed and the data are presented as the mean ± SD. Significance was determined with a Welch's $t$ test with the $P$-value indicated. **(D)** $sbp2^{-/-}$ embryos were injected with single guide RNAs targeting the $secisbp2l$ gene at the single-cell stage. At 3 dpf, larvae were incubated with 375 nM $^{75}$Se for 24 h and lysates from individual larvae were analyzed by phosphorimager analysis. Lane 1 is lysate from an uninjected control embryo. The gel contained trihalo compounds and total protein was imaged under UV light (lower panel).
Source data are available for this figure.

possible genotypes (+/+, +/−, −/−) in a single experiment. Embryos were exposed to 375 nM $^{75}$Se-selenite for 24 h and lysates were analyzed by SDS–PAGE followed by phosphorimager analysis. As expected, Fig 5A shows a high degree of variability in the amount of $^{75}$Se-selenite labeling in progeny as compared with the wild-type

parental strain. The significant reduction of radioactive bands in some samples is suggestive of mendelian inheritance of the edited $sbp2$ allele (25%). To confirm that the loss of selenoprotein expression was a result of Sbp2 loss, we repeated the labeling on bona fide $sbp2^{-/-}$ embryos with verified genotypes. Fig 5B confirms

that the $sbp2^{-/-}$ larvae show a substantial loss of [75]Se-labeled proteins of most molecular weights except the ~68-kD species at all three time points and a 24-kD band in day 3, the former of which likely corresponds to thioredoxin reductase. Overall, the loss of selenoprotein production was not uniform. The most notable change was the loss of a diffuse band at ~45 kD and a pair of bands at ~22/20 kD. These results suggest that Secisbp2l may be supporting selenoprotein synthesis, particularly the 68- and 25-kD species, when Sbp2 is absent. In the case of $secisbp2l^{-/-}$ embryos, no band-specific differences were noted when compared with wild-type embryos in the 3–5 dpf period (Fig 5B, left panel). There was, however, a statistically significant moderate reduction of selenoprotein expression in days 3 and 4 (Fig 5C and D). These data establish that the lack of Sbp2 expression is not sufficient to eliminate selenoprotein production, the baseline level of which is affected by the lack of Secisbp2l only at 3 and 4 dpf.

Interestingly, we noted a significant increase in a ~130-kD Coomassie-stained band in $sbp2^{-/-}$ but not wt or $secisbp2l^{-/-}$ larvae (Fig 5B, lower panel, arrow). This band was most prominent in 3 dpf larvae and not evident in 5 dpf larvae. Mass spectrometric analysis of the excised band revealed that this protein is vitellogenin-1 (vtg1), which is an abundant phosphoprotein present in yolk. As a well-studied biomarker for various types of stress, it provides a clear indication that the loss of $sbp2$ but not $secisbp2l$ initiates a stress pathway that ultimately leads to vtg1 induction. Because vtg1 is known to be induced by estrogens, it is likely that the loss of selenoprotein expression is either directly or indirectly affecting the estrogen synthetic pathway. This is consistent with a previously observed correlation between selenium status, selenoprotein production and estrogen levels in rats (Damdimopoulos et al, 2004).

## Analysis of embryos lacking both Sbp2 and Secisbp2l

To generate a line lacking both Sbp2 and Secisbp2l, we targeted $secisbp2l$ for CRISPR/Cas9 mutagenesis in the $sbp2^{-/-}$ background. We injected 80 embryos alongside the same number of uninjected controls. After 72 h, we collected six embryos for genomic DNA analysis, 24 embryos for metabolic labeling with [75]Se-selenite and 30 embryos each were collected to be raised to study their survivorship. Notably, only one of the injected F0 animals with the $sbp2^{-/-}$ background survived past 14 dpf, and all but one of the genotyped embryos showed evidence of editing at the target site. To assess the effect on selenoprotein production we subjected the injected and control embryos to metabolic labeling with [75]Se-selenite. Fig 5D shows phosphorimager analysis from a subset of injected embryos, all but one of which did not have detectable selenoprotein expression. This result strongly supports the idea that Sbp2 and Secisbp2l work in concert to provide full selenoprotein incorporation. Together, these data reveal that selenoprotein production may not be required for early development in fish but that subsequent survival is limited to an early juvenile stage.

## Quantitative analysis of selenoprotein mRNAs and proteins

In mammalian cells, reduced selenoprotein expression due to limiting selenium or reduced Sbp2 expression is associated with a coordinated but selective reduction in selenoprotein mRNAs (Shetty & Copeland, 2015). Several studies have established that

this response is due to reduced mRNA stability, likely through the nonsense mediated decay pathway (Weiss et al, 1997; Wingler et al, 1999; Sun et al, 2001; Squires et al, 2007; Santesmasses et al, 2019). To examine the response of selenoprotein mRNA levels to selenoprotein reduction due to loss of Sbp2 or Secisbp2l, we performed quantitative RT-PCR at 4 dpf for six selenoprotein mRNAs ($selenow$, $selenop$, $gpx1a$, and $gpx4a$, $selenof$, $trxnrd1$ [note, the zebrafish gene is annotated as $trxnrd3$]) relative to the control mRNA encoding $eef1a1$, which is known to be evenly expressed across 3–5 dpf (McCurley & Callard, 2008). Fig 6A shows that the loss of Sbp2 resulted in an ~5-fold reduction of both $selenop$ and $gpx1a$ mRNAs and a ~2-fold reduction of both $gpx4a$ and $selenow$ but had a much smaller effect on $selenof$, $trxnrd1$, or $trxnrd2$. In the case of $secisbp2l^{-/-}$ the mRNAs that were sensitive to the loss of sbp2 were also sensitive to the loss of secisbp2l but to a much lower extent. We observed ~40% reductions of selenop and gpx4a mRNA, 20% reductions in gpx1a and selenow and either no reduction or increases in txnrd1, txnrd2, and selenof mRNAs. These results establish that the zebrafish system faithfully replicates that observed in mice and mammalian cells with regard to the regulation of selenoprotein mRNA levels under conditions where selenoprotein synthesis is impaired (Fradejas-Villar et al, 2017), and they open up the possibility that secisbp2l is playing a role in regulating selenoprotein mRNA levels.

For quantitative analysis of selenoprotein levels, we examined the proteomes of $sbp2^{-/-}$ and $secisbp2l^{-/-}$ larvae by quantitative mass spectrometry. Four replicate samples at 4 dpf for each strain were lysed in SDS buffer and subjected to in-gel digestion followed by TMT labeling and Nano-LC-MS/MS. As expected, the $sbp2^{-/-}$ proteome was marked by significant decreases in all detectable selenoproteins except for Txnrd1 (Fig 6B). It is also notable, however, that the extent of down-regulation was not uniform with a nearly threefold difference between the low-expressing selenoprotein (Gpx1a) and the highest (Selenop and Selenoj). For Txnrd1, the lack of down-regulation supports the idea that the 68 kD band that is not affected by the loss of Sbp2 is Txrnd1, although it is also possible that what we observed in the mass spec data corresponds to a truncated Sec-free version of Txrnd1, which has been shown to persist in selenium deficient conditions (Legrain et al, 2014). In the case of $secisbp2l^{-/-}$ larvae, we did not detect any reduction of selenoproteins by quantitative mass spectrometry (Fig 6C), although Gpx4a seems to be up-regulated, which was not detectable during [75]Se labeling shown Fig 5. Both datasets contained a discrete set of other proteins that were either up- or down-regulated, which are the subjects of ongoing analysis. Note that Vtg1 was more than 10-fold higher in the $sbp2^{-/-}$ larvae, which is consistent with the band identified in the stained gel in Fig 5B.

## Larvae lacking Sbp2 but not Secisbp2l are sensitive to peroxide stress

Considering the role of selenoproteins in resolving oxidative stress, we sought to determine if the loss of either Secisbp2l or Sbp2 affected sensitivity to chronic oxidative stress. To this end, 0–5 dpf embryos were treated with 2 mM hydrogen peroxide (Fig 7A). Since we started the treatment at 0 dpf, this analysis includes the natural die-off of ~20% of the embryos from 0 to 1 dpf. This is expected as

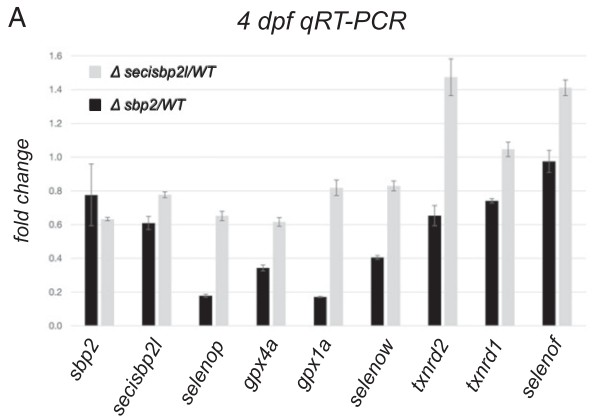

## A — 4 dpf qRT-PCR

*fold change*

Legend: Δ secisbp2l/WT; Δ sbp2/WT

Genes: sbp2, secisbp2l, selenop, gpx4a, gpx1a, selenow, txnrd2, txnrd1, selenof

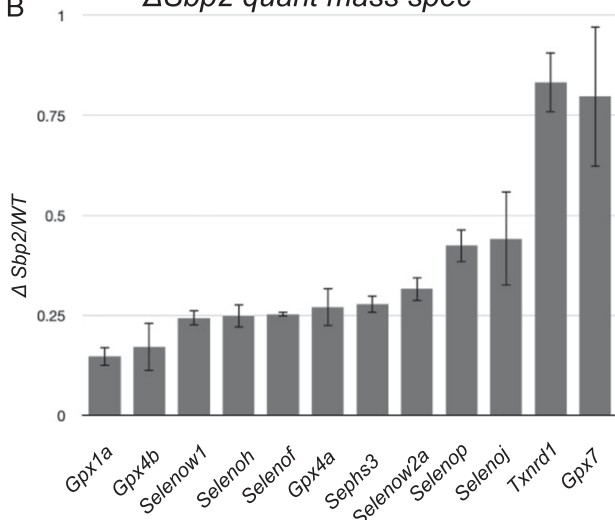

## B — ΔSbp2 quant mass spec

*Δ Sbp2/WT*

Genes: Gpx1a, Gpx4b, Selenow1, Selenoh, Selenof, Gpx4a, Sephs3, Selenow2a, Selenop, Selenoj, Txnrd1, Gpx7

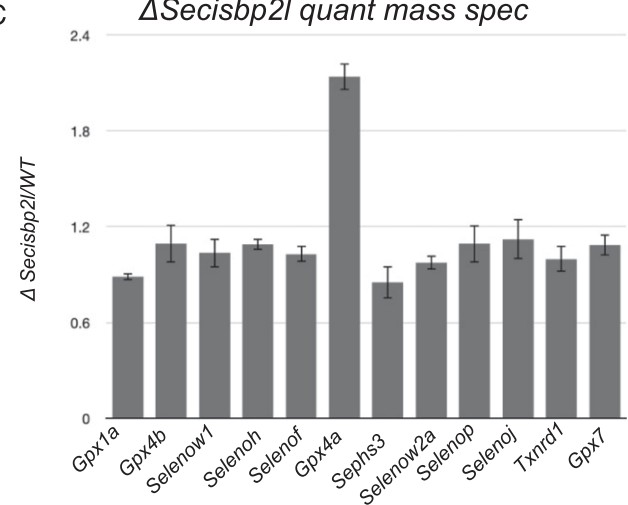

## C — ΔSecisbp2l quant mass spec

*Δ Secisbp2l/WT*

Genes: Gpx1a, Gpx4b, Selenow1, Selenoh, Selenof, Gpx4a, Sephs3, Selenow2a, Selenop, Selenoj, Txnrd1, Gpx7

any unfertilized, or unhealthy embryos die during this time period. Whereas no significant peroxide sensitivity was noted in *secisbp2l*[−/−] larvae or either of the wild-type controls, none of the *sbp2*[−/−] larvae survived past 2 dpf. Interestingly we observed a slightly enhanced survival of *secisbp2l*[−/−] larvae resulting from a lower initial die-off. Overall, these results confirm that the reduction of selenoprotein expression correlates with increased sensitivity to oxidative stress and establishes the *sbp2*[−/−] line as a potential tool for studying hypersensitivity to stress.

Because substantial, albeit selective, selenoprotein synthesis persists in larvae lacking Sbp2, we hypothesize that Secisbp2l is required for "backup" selenoprotein synthesis under stress conditions (e.g., loss of Sbp2). To directly test this hypothesis, we examined the effect of acute peroxide treatment on selenoprotein synthesis in wild-type and *secisbp2l*[−/−] 3–5 dpf larvae. We treated larvae with a sublethal dose of peroxide (0.5 mM) concomitant with [75]Se-selenite labeling for 24 h. Surprisingly, Fig 7B shows that selenoprotein expression was almost completely eliminated by peroxide treatment in the 3 and 4 dpf larvae but was only about 50% reduced at 5 dpf in both the wild-type and *secisbp2l*[−/−] strains, shown quantitatively in Fig 7C. Because the wild-type and mutant response was the same, we did not perform the same experiment on *sbp2*[−/−] animals considering we had previously determined sensitivity to peroxide. These results could not confirm our hypothesis that Secisbp2l is required for a stress response, but they did reveal a developmentally specific response to peroxide that results in dramatic selenoprotein down-regulation or a developmentally specific loss of selenium uptake under these conditions.

## Discussion

We have used the zebrafish system to investigate the roles of the two SECIS-binding proteins in promoting selenocysteine incorporation during development. Overall, our findings have revealed that Sbp2 is the main driver for basal selenoprotein production but that its paralog, Secisbp2l, supports selective selenoprotein expression when Sbp2 is absent. Use of the zebrafish system has allowed us to demonstrate the disparate roles of the two SECIS-binding proteins in the context of embryonic development.

Analysis of *sbp2* and *secisbp2l* genes has revealed that *sbp2* coding sequence in Teleost fish is unique relative to all other vertebrate species. The predicted N-terminal ~80 amino acids are unique in the Clupeocephala supercohort, and there are no conserved motifs that might hint at function. Thus, the N-terminal motifs that are conserved between Sbp2 and Secisbp2l in most vertebrates (Donovan & Copeland, 2009) are present only in the *secisbp2l* gene in Clupeocephala fish. The zebrafish model, then, is unique in providing an opportunity to determine the function for these novel conserved motifs.

**Figure 6. Quantitative analysis of selenoprotein mRNA and protein levels in *sbp2*[−/−] and *secisbp2l*[−/−] larvae.**
**(A)** Total RNA was extracted from 4 days post fertilization larvae and was analyzed by qRT-PCR using primers specific for the genes noted. The data shown are the average ΔΔCT ratios from triplicate analyses using *eef1a* as the control plus/minus SD. **(B)** Total protein lysates from wild-type and *sbp2*[−/−] larvae were subjected to quantitative mass spectrometry. The data shown represent the average mutant/wt peptide count ratio from four replicate samples, plus/minus SD. **(C)** Same as in (B) for the *secisbp2l*[−/−] samples.

https://doi.org/10.26508/lsa.202101291  vol 5 | no 5 | e202101291  

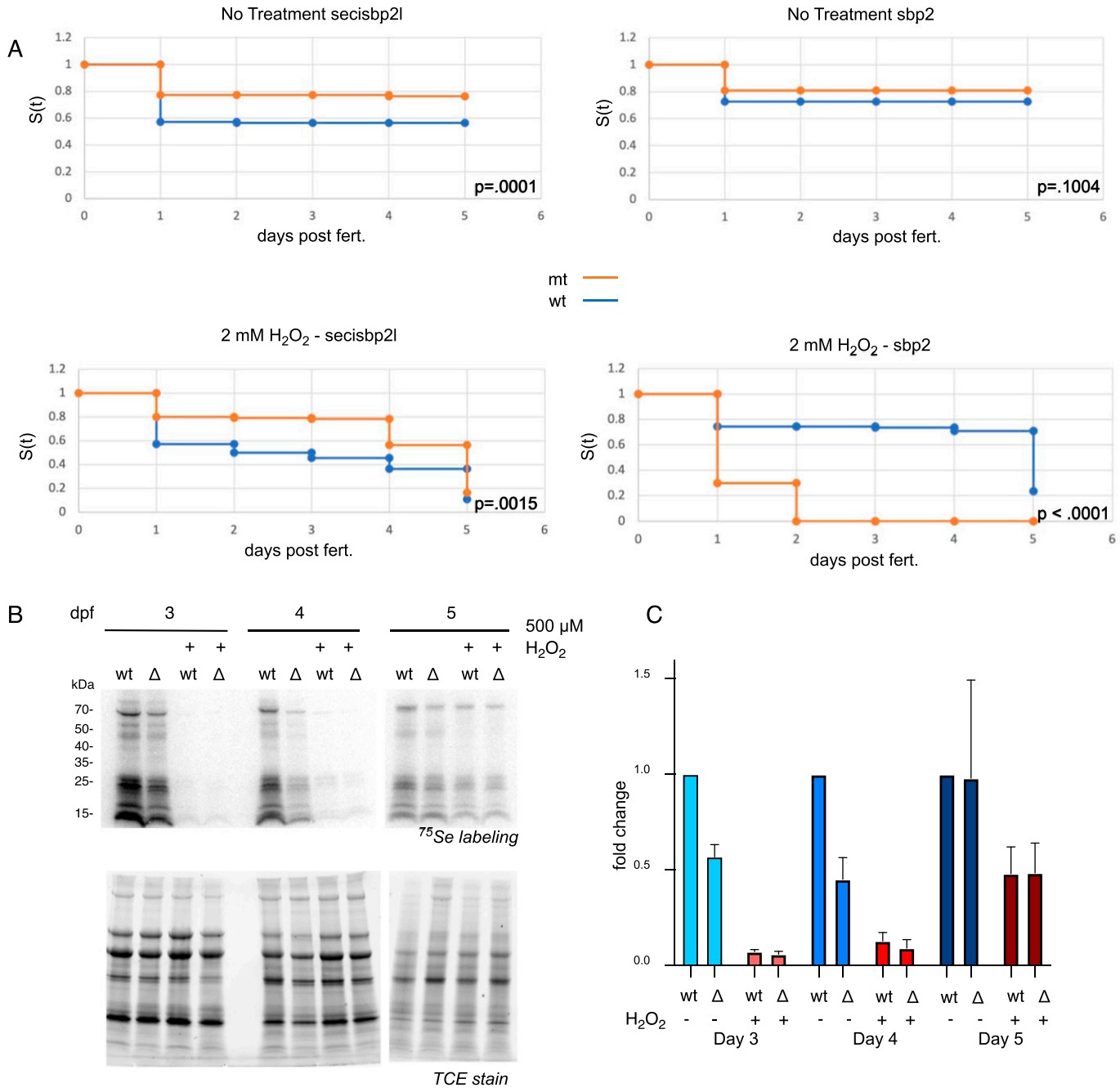

**Figure 7. Mutant *sbp2*<sup>−/−</sup> larvae and general selenoprotein production are sensitive to peroxide treatment.**
**(A)** Wild-type (blue) or mutant (orange) larvae (n = 110) were raised in embryo medium with or without 2 mM hydrogen peroxide for 5 d (water was changed daily). Survival was analyzed by the Kaplan-Meier method and *P*-values were determined by the log-rank test. **(B)** 3–5 days post fertilization wild-type (wt) or secisbp2l<sup>−/−</sup> (Δ) larvae were incubated with 375 nM <sup>75</sup>Se with or without 500 *μ*M hydrogen peroxide for 24 h. Radiolabeled proteins were analyzed by SDS–PAGE followed by phosphorimaging (top panels) and total protein is shown as fluorescence imaging of TCE-stained gels (bottom panel). **(C)** Quantitative analysis of selenoprotein production in the presence or absence of peroxide. **(B)** <sup>75</sup>Se incorporation was determined by quantitation of the phosphorimages shown in (B). Data were normalized to wild-type untreated samples and shown as the average of three experiments plus/minus SD.

Using existing RNA-seq data across zebrafish development (White et al, 2017), we found that selenoprotein mRNAs are expressed throughout development but at highly individualized levels and times. The burst of expression for *gpx4a/b* and *selenop* after ~10 hpf is striking in that it is not accompanied by a concomitant increase in any of the mRNAs encoding selenoprotein synthetic factors, including *sbp2 and secisbp2l*. This may indicate the presence of a large pool of long-lived maternal protein, but further work will be required to determine the significance of this apparent disconnect between the levels of selenoprotein synthetic

factor mRNAs and selenoprotein mRNAs. Interestingly, the slight offset of *gpx4a* versus *gpx4b* expression is consistent with the non-overlapping spatial restriction of gpx4a to the yolk sac, liver, and kidney versus the brain and myotomes for *gpx4b*, which was observed as part of a broad analysis of spatial and temporal selenoprotein mRNA expression in zebrafish embryos (Thisse et al, 2003). This highlights the diverse array of selenoprotein functions during development and stresses the importance of future studies directed toward the coordinated regulation of the factors that are required for selenoprotein production.

Our analysis of *sbp2*⁻/⁻ fish was surprising in that zebrafish larvae are able to survive this condition, whereas mouse embryos die early during gastrulation (Seeher et al, 2014). Although our immunoblot analysis could not definitely show a complete lack of Sbp2 because of the presence of cross-reacting species, sequencing at the genomic and transcript level confirmed that wt *sbp2* is not detectable. As such, it is likely that the relatively low oxygen exposure of fish eliminates the requirement for a full complement of selenoproteins during early development. This point was emphasized by our finding that injecting anti-*secisbp2l* CRISPR sgRNA was sufficient to completely eliminate selenoprotein synthesis. This is the first demonstration that Sec incorporation is fully dependent on SECIS-binding proteins because all prior works have focused solely on Sbp2. The fact that these embryos did not survive past day 14 pinpoints a surprisingly late time in development that is sensitive to the loss of selenoproteins. Although it is not possible to pinpoint which selenoproteins might be required for survival, considering the central role of GPX4 in responding to ferroptotic damage and its ability to allow survival of mouse embryos that lack other selenoproteins (Ingold et al, 2018), it is a good candidate for playing a similarly key role in fish.

We found that selenoprotein production was significantly but selectively down-regulated in *sbp2*⁻/⁻ but not in *secisbp2l*⁻/⁻ larvae. The primary exception to down-regulation is a 68-kD selenium-labeled protein that we predict corresponds to thioredoxin reductase (Txnrd1). This leads to the key mechanistic question of whether Secisbp2l preferentially binds certain selenoprotein mRNAs. Considering that ablating Secisbp2l led to a modest reduction of all selenoproteins, it is more likely that selective expression in the *sbp2*⁻/⁻ background is mainly a result of the altered mRNA levels in that strain. Indeed, qPCR and quantitative mass spec analysis revealed a good correlation between the reduced mRNA and protein levels for *gpx4*, *gpx1*, and *selenow*. However, in the case of *selenof*, substantial loss of protein was not accompanied by a change in mRNA levels. In the case of *secisbp2l* ablation, most mRNA levels were within 60–80% of wild-type with the exception of *selenof* and *txnrd2*, which showed a ~40% increase that was not apparent at the protein level. In addition, Gpx4a protein levels were found to be more than twofold higher in the *secisbp2l*⁻/⁻ strain but mRNA levels were reduced to 60% of wild-type. These results are consistent with substantial translation regulation that might be expected from the loss of a 3′ UTR-binding protein. Targeted mechanistic studies of these cases may reveal novel regulatory mechanisms that are unique to Secisbp2l.

The fact that selenoproteins are still made in the absence of Sbp2 strongly implicates Secisbp2l as a factor that supports "survival" selenoprotein production under conditions where Sbp2 may be limiting. Prior work in mammalian cells indicated that

oxidative stress resulted in glutathionylation and altered subcellular localization of Sbp2 from the cytoplasmic to the nuclear compartment, thus leading to reduced selenoprotein synthesis (Papp et al, 2006). Although this effect seen in cultured cells may not be related to what we observed in zebrafish, it is consistent with the idea that the primary role of Secisbp2l in the context of Sec incorporation may be to shift selenoprotein synthesis to a survival mode during stress. However, we found *secisbp2l*⁻/⁻ animals to be no more sensitive to peroxide stress than wild-type controls. This was in contrast to *sbp2*⁻/⁻ animals where the larvae did not survive past 2 dpf in the presence of peroxide. We also found that peroxide treatment almost completely eliminates selenoprotein expression in 3 to 4 dpf but not 5 dpf embryos. The resistance at 5 dpf serves as an important control in reducing the likelihood that peroxide is preventing selenium uptake. This was an unexpected result considering that other studies have demonstrated reduced selenoprotein expression in the presence of peroxide, but this dramatic temporally regulated response has not previously been reported. This result also raises a question about why only *sbp2*⁻/⁻ larvae were sensitive to 2 mM peroxide. If the sensitivity was due to the lack of selenoproteins and peroxide is eliminating all selenoprotein production, then both lines should have been sensitive. The continued use of zebrafish will allow mechanistic evaluation of this potentially novel stress pathway as further dissection of the response to peroxide stress will be required to determine the basis for this striking regulation of selenoprotein expression.

One clue about the type of stress resulting from the loss of Sbp2 may lie in our observation that *sbp2*⁻/⁻ animals showed substantial up-regulation of vitellogenin (Vtg1) which has previously been noted to be up-regulated in stress conditions, specifically as a target of environmental toxins that are estrogen receptor agonists (Hultman et al, 2017). This confirms the idea that the loss of Sbp2 causes a specific type of stress that is likely caused by the loss of one or more selenoproteins. Future work with these zebrafish lines will allow the Identification of precisely what type of stress might lead to the condition where Secisbp2l driven Sec incorporation is required, ultimately providing mechanistic insight into why Secisbp2l expression is positively correlated with protection from lung cancer.

# Materials and Methods

### Zebrafish husbandry

All fish were maintained in a flow-through system with a light/dark cycle of 14/10 h according to standard procedures (Mullins et al, 1994). Embryos and larvae (0–5) dpf were grown at 28°C in embryonic medium (0.005% Instant Ocean and 0.1% Methylene Blue). We performed all experiments involving fish according to animal protocols that were approved by Rutgers, The State University of New Jersey.

### Generation of Secisbp2l and Sbp2 knockout zebrafish

Ablation of the *secisbp2l* gene was performed by Knudra, now InVivo Biosystems. An in vitro assembled CRISPR–Cas9 RNP complex

**Table 1.** List of oligonucleotide sequences used in this study.

| Oligonucleotide description | Sequence |
|---|---|
| *secisbp2l* single guide RNA | UAAGCUGGUGUCCCUGACGG |
| crRNA | UCCCGGUCAGCU |
| *secisbp2l* genotyping fwd | GGCTGTTCAATAAGCTGGTGTC |
| *secisbp2l* genotyping rev | AGATGACGCTGCAGAAGGAG |
| *sbp2* genotyping fwd | CTTAGGTGGTCTAGATGAGGC |
| *sbp2* genotyping rev | TCTTCCTGTAGATCCTCCTGCG |
| *sbp2* cloning fwd | ATGGAGAATCATTCAAAAAGAGCTC |
| *sbp2* cloning rev | ACTTCCTTCATCGTCCAGAAGCTG |
| *secisbp2l* cloning fwd | GAGGAAGGATGTAAAGCTCTCTGC |
| *secisbp2l* cloning rev | GATCTGCCGCTGGTTCAGTG |

(consisting of a sgRNA [Table 1], and recombinant Cas9) was microinjected into TU zebrafish embryos at the single-cell stage. Injected embryos were shipped to Rutgers and raised to the juvenile stage before genotyping (see below). Ablation of the *sbp2* gene was performed in-house via microinjection into EK zebrafish embryos at the single-cell stage using an in vitro assembled CRISPR–Cas9 RNP complex (consisting of a crRNA, tracrRNA, and recombinant Cas9). Alt-R CRISPR–Cas9 crRNA (IDT) and Alt-R CRISPR–Cas9 tracrRNA, ATTO 550 (IDT) were diluted to a final concentration of 3 µM each in IDTE buffer (10 mM Tris and 0.1 mM EDTA, pH 8.0), heated to 95°C for 5 min, and then cooled to room temperature. In a separate tube, Alt-R S.p. Cas9 Nuclease V3 (IDT) was diluted to a final concentration of 1.5 µg/µl in Cas 9 working buffer (20 mM Hepes and 150 mM KCl, pH 7.5). Equal volumes of each solution were incubated together at 37°C for 10 min before being cooled to room temperature then injected into single-cell embryos (Wierson et al, 2019).

### Generation of Secisbp2l and Sbp2 double knockout zebrafish

Ablation of both *sbp2* and *secisbp2l* gene was performed via microinjection of in vitro assembled CRISPR–Cas9 RNP complex (consisting of a crRNA, tracrRNA, and recombinant Cas9) against the *secisbp2l* gene into $sbp2^{-/-}$ zebrafish embryos at the single-cell stage. Alt-R CRISPR–Cas9 crRNA (IDT) and Alt-R CRISPR–Cas9 tracrRNA, ATTO 550 (IDT) were diluted to a final concentration of 3 µM each in IDTE buffer (10 mM Tris and 0.1 mM EDTA, pH 8.0), heated to 95°C for 5 min, and then cooled to room temperature. In a separate tube, Alt-R S.p. Cas9 Nuclease V3 (IDT) was diluted to a final concentration of 1.5 µg/µl in Cas 9 working buffer (20 mM Hepes and 150 mM KCl, pH 7.5). Equal volumes of each solution were incubated together at 37°C for 10 min before being cooled to room temperature then injected into single-cell embryos (Wierson et al, 2019).

### Genotyping

For genotyping, DNA was extracted from juvenile tail fragments in 1× PCR buffer (10 mM Tris HCl, pH 8.3, 50 mM KCl, 0.3% IGEPAL, and 0.3% Tween 20), which were lysed at 95°C for 10 min. Samples were cooled to 55°C and PCR grade Proteinase K (Roche) was added to a concentration of 1 µg/µl, and samples were incubated for 16 h at 55°C. Proteinase K was then heat inactivated for 10 min at 95°C and a fragment surrounding the target site was generated by PCR. PCR products were gel purified using the PureLink Quick Gel Extraction Kit (Invitrogen) and sequenced. F0 adults with detectable editing were crossed to wild-type EK zebrafish adults. F1 generation fish that had frameshift mutations were isolated and an allele with a 26-bp exonic insertion in the *secisbp2l* gene was chosen for further analysis, whereas a 5-bp exonic deletion in the *sbp2* gene was chosen. F1 interbreeding created F2 fish homozygous knockout, heterozygous, and wild-type siblings for each allele. F2 fish homozygous for the 26 bp allele ($secisbp2l^{-/-}$) and homozygous wt ($secisbp2l^{+/+}$) for the *secisbp2l* gene were used in experiments. In the case of SBP2, F2 fish homozygous for the 5 bp allele ($sbp2^{-/-}$) and homozygous wt ($sbp2^{+/+}$) for the *sbp2* gene were used for all experiments in this report. The primers used for genotyping single and double knockout animals are shown in Table 1.

### RNA extraction and qR-TPCR analysis

Pools of at least 15 embryos were euthanized and placed in a 1.5 ml conical tube and all embryo medium was withdrawn. Embryos were then immediately flash frozen on dry ice. Samples were then thawed on ice and the manufacturer's protocol for solid sample TRIzol Reagent (Ambion) extraction was followed. To remove genomic DNA, RNA cleanup protocol of RNeasy Mini Kit (QIAGEN) with DNase treatment was used. cDNA was generated from total RNA using the Superscript IV Reverse Transcriptase (Thermo Fisher Scientific) following the manufacturer's protocol using poly(A) priming and 200 ng of total RNA. 10 ng/rxn of cDNA was then used in qRT-PCR using *Power*SYBR Green PCR Master Mix (Applied Biosystems) with QuantStudio 5 Real-Time PCR System (Thermo Fisher Scientific). The total reaction volume was 20 µl with 5 µl of 1:5 diluted RT reaction. Working concentration of the primers in the reaction was 0.25 µM. Thermal cycling conditions were 95°C for 10 min followed by 40 cycles of 95°C for 15 s, 60°C for 1 min. Melt curve analysis was performed for each sample to ensure a single amplification product. Samples were analyzed in triplicate for both the reference gene and the target gene. Quantitation was performed using the comparative ΔΔCt method. We used *eef1a* as the normalizer and the calibrator sample was from each wild-type background. Primers in the acceptable efficiency range (90–110%) were determined using the standard curve method.

### secisbp2l and sbp2 cDNA cloning

Coding sequence for *sbp2* was PCR-amplified from 0 to 5 dpf pooled cDNA using the forward and reverse primers shown in Table 1 and then TA cloned into pcDNA3.1 V5/His Topo (Thermo Fisher Scientific). For *secisbp2l*, most of the predicted coding region was amplified as overlapping fragments from 0 to 5 dpf pooled cDNA. Specifically the sequence obtained corresponded to the start of exon 2 (22 nt downstream from the predicted start) to the stop codon. The 5′ end of the cDNA was added as a synthesized fragment based on a multiple sequence alignment with other fish sequences. Note that the *sbp2* gene in goldfish is annotated as LOC113080603 and the corresponding mRNAS are incorrectly annotated as *secisbp2l*.

## Metabolic labeling with $^{75}$Se

Embryos were placed in multiwell plates at a density of 83 $\mu$l per embryo in labeling medium (embryo medium containing 0.5% DMSO (vol/vol) and 375 nM $^{75}$Se [~500 Ci/g; University of Missouri Reactor]). Embryos were incubated at 28°C for 24 h. After incubation, embryos were transferred to wash vessels at a density of about 10 ml per embryo for 10 min. Four wash cycles were sufficient to remove background of $^{75}$Se label. Embryos were immediately frozen on dry ice and stored at –80°C until used for protein extraction.

## Protein extraction of zebrafish embryos

Frozen aliquots of 1–4 fish were thawed on ice and 15–30 $\mu$l of extraction buffer (63 mM Tris–HCl, pH 6.8, 10% glycerol [vol/vol], 3.5% SDS [wt/vol], and 5.0% BME [vol/vol]) was added to the embryos before the fish were thoroughly homogenized with a pestle attached to a power drill run at ~700 RPM. The samples were kept on ice, whereas three 10-s pulses were delivered, separated by 10-s resting intervals. Samples were briefly centrifuged then heated to 85°C for 10 min and vortexed at room temperature halfway through heating. Samples were centrifuged for 20 min at 14,000$g$ and the supernatant was isolated. Bromophenol blue was added to a concentration of 0.1% (wt/vol) to samples. Samples were then analyzed by SDS–PAGE and the gels dried and exposed to PhosphorImager screens (GE Healthcare). Total protein was monitored by Coomassie blue staining.

## Immunoblotting

SDS–PAGE was performed on lysates and transferred to nitrocellulose via wet transfer (Bio-Rad). SECISBP2 Polyclonal antibody (12798-1-AP; ProteinTech) was used at a 1:1,000 dilution, and Beta Tubulin Monoclonal antibody (66240-1-Ig; ProteinTech) at 1:20,000 dilution. Horseradish peroxidase–conjugated anti-rabbit IgG secondary (65-6120; Invitrogen) was used at a dilution of 1:40,000. Blots were developed using the SuperSignal West Femto kit (Pierce) or fluorescence imaging according to the manufacturer's protocol with Amersham Imager 600 (GE Healthcare).

## Hydrogen peroxide treatment

For the toxicity assay, 0 dpf embryos were transferred to 10 cm petri dishes containing 40 ml of labeling medium either with or without 2 mM $H_2O_2$ (Sigma-Aldrich) at a density of 1 ml embryo medium per embryo and kept at 28°C. Toxicity was assessed every 24 h under a light microscope, checking for the evidence of a heartbeat. Embryos were removed from the dish when heartbeat was not detected, and medium was changed every 24 h from the time of initial treatment. For labeling during peroxide treatment, 2–4 dpf embryos were transferred to a 12-well plate containing 1 ml of labeling medium with or without 0.5 mM $H_2O_2$ per well. After 24 h of treatment, larvae were washed then flash frozen on dry ice and processed for SDS–PAGE and phosphorimager analysis as described above.

## Quantitative mass spectrometry

4 dpf larval extracts were prepared (4 larvae per tube) as described above. Mass spectrometry was performed at the Biological Mass Spectrometry facility at Rutgers Robert Wood Johnson Medical School. The entire sample was loaded onto an SDS–PAGE gel and run just enough to enter the gel. The gel was stained with Coomassie blue and all stained material was recovered as a gel slice. The gel slices were incubated at 60°C for 30 min with 10 mM DTT. After cooling to room temperature, 20 mM iodoacetamide were added and kept in the dark for 1 h to block free cysteine. The samples were digested by trypsin at 1:50 (w:w, trypsin:sample) and incubated at 37°C overnight. The digested peptides were extracted and dried under vacuum and washed with 50% acetonitrile to pH neutral. The digested peptides were labeled with Thermo TMTpro (Lot #: UI292951) following the manufacturer's protocol. Labeled samples were pooled at 1:1 ratio for a small volume and analyzed with LC–MS/MS to get normalization factor. The labeling efficiency is 97.8%. The labeled samples were then pooled at 1:1 ratio for all the channels. The pooled samples were dried and desalted with SPEC C18 (WAT054960; Varian). The desalted samples were fractionated using Agilent 1,100 series. The samples were solubilized in 200 $\mu$l of 20 mM ammonium formate, pH 10, and injected onto an Xbridge column (C18 3.5 $\mu$m 2.1 × 150 mm; Waters) using a linear gradient of 1% B/min from 2 to 45% of B (buffer A: 20 mM ammonium, pH 10, B: 20 mM ammonium in 90% acetonitrile, pH 10). 1-min fractions were collected and dried.

Nano-LC-MSMS was performed using a Dionex rapid-separation liquid chromatography system interfaced with a Eclipse (Thermo Fisher Scientific). Selected desalted fractions 28–45 were loaded onto a Acclaim PepMap 100 trap column (75 $\mu$m × 2 cm; Thermo Fisher Scientific) and washed with Buffer A (0.1% trifluoroacetic acid) for 5 min with a flow rate of 5 $\mu$l/min. The trap was brought in-line with the nano analytical column (nanoEase, MZ peptide BEH C18, 130A, 1.7 $\mu$m, 75 $\mu$m × 20 cm; Waters) with flow rate of 300 nl/min with a multi step gradient (4–15% buffer B [0.16% formic acid and 80% acetonitrile] in 20 min, then 15–25% B in 40 min, followed by 25–50% B in 30 min).

The scan sequence began with an MS1 spectrum (Orbitrap analysis, resolution 120,000, scan range from 350 to 1,600 Th, automatic gain control [AGC] target $1 \times 10^6$, maximum injection time 100 ms). For SPS3, MSMS analysis consisted of collision-induced dissociation (CID), quadrupole ion trap analysis, AGC $2 \times 10^4$, (normalized collision energy 35, maximum injection time 55 ms), and isolation window at 0.7. After acquisition of each MS2 spectrum, we collected an MS3 spectrum in which 10 MS2 fragment ions are captured in the MS3 precursor population using isolation waveforms with multiple frequency notches. MS3 precursors were fragmented by HCD and analyzed using the Orbitrap (normalized collision energy 55, AGC $1.5 \times 10^5$, maximum injection time 150 ms, resolution was 50,000 at 400 Th scan range 100–500). The whole cycle is repeated for 3 s before repeat from an MS1 spectrum. Dynamic exclusion of 1 repeat and duration of 60 s was used to reduce the repeat sampling of peptides.

LC-MSMS data were analyzed with Proteome Discoverer 2.4 (Thermo Fisher Scientific) with a sequence search engine against uniprot human and a database consisting of common laboratory

contaminants. The MS mass tolerance was set at ± 10 ppm, MSMS mass tolerance was set at ± 0.4 D. TMTpro on K and N-terminus of peptides and carbamidomethyl on cysteine was set as static modification. Methionine oxidation, protein N-terminal acetylation, N-terminal methionine loss, or N-terminal methionine loss plus acetylation were set as dynamic modifications. Percolator was used for results validation. Concatenated reverse database was used for target-Decory strategy. High confidence for protein and peptides were defined as false discovery rate (FDR) < 0.01, medium confidence was defined as FDR < 0.05.

For reporter ion quantification, reporter abundance was set to use signal/noise ratio (S/N) if all spectrum files have S/N values. Otherwise, intensities were used. Quan value was corrected for isotopic impurity of reporter ions. Co-isolation threshold was set at 50%. Average reporter S/N threshold was set at 10. SPS mass matches % threshold was set at 65%. Protein abundance of each channel was calculated using summed S/N of all unique+razor peptides. The abundance was further normalized to the summed abundance value for each channel over all peptides identified within a file.

Protein abundance ratio was calculated based on protein abundance. Pairwise comparison was done using t test. P-value was further corrected by Benjamini–Hochberg method for FDR. Volcano plots were generated by VolcanoseR (Goedhart & Luijsterburg, 2020) and gene ontology was performed by the Molecular and Genomics Informatics Core at Rutgers New Jersey Medical School.

## Data Availability

Quantitative mass spectrometry data were deposited to the mass spectrometry interactive virtual Environment (MassIVE) database as doi:10.25345/C5VP39.

## Supplementary Information

## Acknowledgements

We thank Sumangala Shetty for helpful advice and Haiyan Zheng for help with mass spectrometry data analysis. We also thank Joseph Kramer and Katie Flaherty for assistance with techniques. This work was supported by a grant from the National Institutes of General Medical Sciences, R01GM077073 (PR Copeland).

## Author Contributions

NT Kiledjian: conceptualization, data curation, formal analysis, validation, investigation, visualization, methodology, project administration, and writing—review and editing.
R Shah: conceptualization, data curation, formal analysis, validation, visualization, methodology, and writing—review and editing.
MB Vetick: conceptualization, validation, investigation, methodology, and writing—review and editing.
PR Copeland: conceptualization, data curation, formal analysis, supervision, funding acquisition, project administration, and writing—original draft, review, and editing.

## Conflict of Interest Statement

The authors declare that they have no conflict of interest.

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
