## [Reviewer comments · Life Science Alliance]

Life Science Alliance

The expression of essential selenoproteins during development requires SECIS binding protein 2-like

Nora Kiledjian, Rushvi Shah, Michael Vetick, and Paul Copeland

DOI: <https://doi.org/10.26508/lsa.202101291>

Corresponding author(s): Paul Copeland, Rutgers, The State University of New Jersey

Review Timeline:

Submission Date:	2021-11-08
Editorial Decision:	2021-12-17
Revision Received:	2021-12-23
Editorial Decision:	2022-01-11
Revision Received:	2022-01-20
Accepted:	2022-01-21

Scientific Editor: Novella Guidi

Transaction Report:

December 17, 2021

Re: Life Science Alliance manuscript #LSA-2021-01291-T

Dr. Paul Copeland
Rutgers-Robert Wood Johnson Medical School
Biochemistry and Molecular Biology
675 Hoes Ln
Piscataway, NJ 08854

Dear Dr. Copeland,

Thank you for submitting your manuscript entitled "The expression of essential selenoproteins during development requires SECIS binding protein 2-like" to Life Science Alliance. The manuscript was assessed by expert reviewers, whose comments are appended to this letter. We, thus, encourage you to submit a revised version of the manuscript back to LSA that responds to all of the reviewers' points.

Thank you for this interesting contribution to Life Science Alliance. We are looking forward to receiving your revised manuscript.

Sincerely,

B. MANUSCRIPT ORGANIZATION AND FORMATTING:

Reviewer #1 (Comments to the Authors (Required)):

In this manuscript entitled "The expression of essential selenoproteins during zebrafish development requires SECIS binding protein 2-like." The authors investigated the role of SECIS binding protein 2-like (Secisbp2l) for the expression of selenoproteins in zebrafish. This study includes interesting and new observations, while several descriptions were inadequate and included overinterpretation. My major comments are as follows:

Major comments:

1. It appears that the role of Secisbp2l is supplementary for SBP2, and under stress experiment, Secisbp2l mutant did not show remarkable phenotype as shown in Fig. 7A. However, the authors stated as "We propose a model where Secisbp2l is required to promote essential selenoprotein synthesis during stress." This statement includes overestimations of results.
2. In the introduction, the description of Secisbp2l is less. The authors should describe the previous studies related to this gene and the character of Secisbp2l similar to SBP2 here.
3. The quality of Fig.1 is low and the description is insufficient. In addition, to understand this figure smoothly, the description of Secisbp2l in the introduction is helpful for readers.
4. The quality of Fig. 2 is also low and hard to see. The authors should improve this.
5. Again, the quality of Fig. 4 is low, and the arrow did not indicate a specific band precisely. In addition, the term "Secisbp2l" is described as "Secisbp2". Which is correct? The authors should check this point in the main text (for example, the title on page 16, "secisbp2^{-/-} embryos").
6. The authors showed the high variability of selenoprotein levels in Fig. 5A. The meaning of this figure is unclear. Were these bands from sbp2^{-/-} or includes sbp^{+/+} and +/-? What is the purpose of this figure? Is it correlated with other phenotypes?
7. The authors investigated the role of the Secisbp2l gene in zebrafish. Of course, this experimental system has some merit, while the experiments using cultured cells might help to understand the precise role of Secisbp2l for the synthesis of selenoproteins.
8. The interpretation of Fig. 7 is wrong. The authors should think about the meaning of this experiment precisely.
9. As the authors mentioned, the effects of vitamin E and deferoxamine will be interesting.
10. In the discussion (page 23), the meaning of the chapter "The caveat to the idea that Secis..." was unclear. The authors should consider the obtained results precisely.

Reviewer #2 (Comments to the Authors (Required)):

In this study, Kiledjian and co-authors investigated the relative role of the two SECIS binding proteins Sbp2 and its paralogue Secisbp2l in selenoproteins synthesis during zebrafish embryo development. Sbp2 is one well-known key player in selenocysteine insertion mechanism, but the function of the Secisbp2l, conserved in vertebrates, remains elusive. Both genes were mutated using a CRISPR strategy, either alone or in combination, and the outcomes on the selenoproteome composition were monitored using ⁷⁵Se labeling. It shows that each gene deleted individually had limited incidence on selenoprotein expression, indicating a redundant function for the two genes. Only deletion of both genes prevented selenoprotein synthesis, leading to embryonic lethality at 14 days post fertilization. Finally, higher oxidative stress sensitivity was observed in sbp2 depleted embryos, but not in the secisbp2l depleted ones. This study addressed the interesting and sometime controversial question of paralogous genes function, here in the context of selenoprotein translation and expression during zebrafish development.

The topic of this work is interesting and the manuscript is well written. However, it is several cavities raising important questions. I hope the following comments will help to clarify some of these questions and improve the impact for a large audience.

It is a problem in the last sentence of the summary: "Embryos lacking Sbp2 are sensitive to oxidative stress and express the stress marker vtg1. We propose a model where Secisbp2l is required to promote essential selenoprotein synthesis during stress". This last sentence is confusing as one will understand that Secisbp2l is required to promote essential selenoprotein synthesis during oxidative stress, which is in contradiction with the results obtained. From the results part, it is more likely that it

is postulated that Secisbp2l is required to promote essential selenoprotein synthesis in absence of Sbp2. Although, this point is not clear at all (see below).

"Additionally the muscle-specific selenoprotein SELENON was reported to be required for normal calcium flux in zebrafish embryos". This information lacks references. In addition, a comprehensive analysis of selenoprotein expression patterns in zebrafish embryos was reported (Thisse et al., 2003, Mechanism of Development-Expression Pattern, 3, 525-532). Data provided in this publication might be helpful to analyze Sbp2 and Secisbp2l deletion mutants. In addition, comparison between mined data from RNA-seq and the described selenoprotein expression patterns will be interesting.

The secisbp2l^{-/-} mutants were generated on a mixed TU/EK genetic background, as the sbp2^{-/-} and the double secisbp2l and sbp2^{-/-} mutants were generated on a pure EK background. The possible differences due to this genetic heterogeneity need to be mentioned and commented.

The abbreviation dpf is not introduced.

Mat & Meth, page 9 "Hydrogen Peroxide Treatment" - "at a density of 1000ul of embryo medium/embryo". The volume unit is not clear.

Mat & Meth, page 10 "The digested peptides were labeled with Thermo TMTpro (Lot #: UI292951) followed by the manufacturer's protocol". This sentence sounds strange.

Figure 1 is too small; it cannot be read properly. What are the two gene diagrams displayed in purple and green (not mentioned in the legend) In addition, the significance of the multiple lower arrows displayed at the bottom is not clear since it is mentioned in the text that a single guide RNA targeting position 2325-2344 for secisbp2l or position 1955-1977 for spb2, was used.

Figure 2, a clustering of the different genes according to their expression pattern rather than displaying them by alphabetic order might help to get a clearer analyzes of the data and their categorization into different classes.

Figure 5 needs to be properly described including top and lower panel for each case.

Figure 5A, it is mentioned in the text "The significant reduction of radioactive bands in some samples corresponds to mendelian inheritance of the edited sbp2 allele (25%)". Actually on the figure, 5 lanes over the 16 analyzed showed reduced intensity; it is difficult from such sample size to deduce a statistically significant mendelian distribution.

"In the case of secisbp2l^{-/-} embryos, no band-specific differences were noted when compared to wild-type embryos in the 3-5 dpf period (Figure 5C)". Actually it is Figure 5B, left panel.

Figure 5D, which panel is the 75Se labeling experiment? Does the Commassie staining appear in grey scale for this experiment? In addition this CRISPR/Cas9-mediated secisbp2l mutagenesis in the sbp2^{-/-} background raise questions: for the double knock-out, it appears that most CRISPR injected animals (F0) were deleted with both secisbp2l alleles, and these embryos displayed no selenoprotein synthesis and died before 14 dpf. However, few animals survived. Were those animals depleted at a heterozygous state? Inbreeding of these heterozygous animals might help to prove and not just support the importance of both genes for full selenocysteine incorporation (not selenoprotein incorporation).

Figure 6A, the unexpected increased expression in txnrd2 and selenof mRNAs expression in the Δsecisbp2l context requires comments. Similarly, in Figure 6C, level of GPx4a detected by mass spectrometry analysis is not in agreement with the mRNA level (Figure6A). How to interpret this result?

Figure 7, it is a problem in panel names displayed on the figure, compared to the legend text. In addition, it is an exchange between colors used for the WT and mutant embryos.

Result of the last experiment combining treatment sublethal dose of peroxide and 75Se labeling are puzzling: complete removal of selenoprotein expression in 3 and 4 dpf embryos was not dependent on the genotype (no differences between WT and Δ). It is more likely that the oxidative treatment prevent Se absorption (it is still reduced selenoprotein expression at 5 dpf) or selenocysteine insertion into proteins. How is this relevant to the hypothesis that Secisbp2l is required for "backup" selenoprotein synthesis under loss of Sbp2? To my opinion, this hypothesis should be tested in the sbp2^{-/-} background? This part is very confusing and the data obtained are overinterpreted.

The discussion overall is interesting.

Altogether, the study presented here is of interest and most results are well analyzed. However, the manuscript can still be improved before publication.

Reviewer #3 (Comments to the Authors (Required)):

The manuscript of Kiledjan et al. aims at understanding the relative importance of two related genes, *spb2* and *secisbp2l*, involved in selenoprotein synthesis using the zebrafish model and the CRISPR-Cas9 approach. They show that *spb2*^{-/-} embryos are still able to produce some selenoproteins whose expression is totally suppressed by the invalidation of the two genes. Furthermore, they studied the sensitivity to oxidative stress of fish invalidated for one or the other of the two genes and showed a differential sensitivity between the two mutants. Their beautiful work therefore sheds new light on the role of the *secisbp2l* gene, which is conserved in vertebrates and whose role is not well understood.

Minor remarks:

- There is no reference in the first section of the introduction about the specific selenoprotein synthesis machinery. It would be good to cite some of them.
- "The only annotated sequence that contains a predicted sequence upstream of the conserved domains was also found in goldfish. However, the *secisbp2* gene in goldfish is annotated as LOC113080603 and the corresponding mRNAs are incorrectly annotated as *secisbp2l*. Interestingly, the N-terminal ~80 amino acids of the predicted goldfish *Sbp2* are only found in the Clupeocephala supercohort of teleost fish, which includes the cyprinidae (zebrafish and goldfish). The N-terminal sequence that was previously found in both *Sbp2* and *Secisbp2l* in most species (Donovan and Copeland 2009) is only present in Clupeocephala *Secisbp2l* but not in *Sbp2*." This part is difficult to understand. It should be rewritten in a clearer manner.
- Nowhere is it mentioned whether the *spb2* and *secisbp2l* genes have paralogs, as many other genes, including some selenoprotein genes, were duplicated during the third round of duplication (3R) that specifically took place in ray-finned 350 mya ago.
- In Figure 1, there are more black arrows than Crisper targets. What do the other arrows indicate?
- Regarding figure 2, what do the numbers and the term TPM (in the text) correspond to and how was the color code established?
- Figure 3, if it is the proteins and their different domains that are represented on the top diagrams in A and B the names on the left should not be in italics. Otherwise the diagrams represent what exactly? Also, it would be more convenient for the comprehension to indicate the location of the premature stop codon in each case.
- "For *Sbp2* analysis we were able to use an affinity purified commercial antibody raised against a C-terminal portion of the human protein (aa 506 - 854) which is 79% identical in the conserved L7Ae RNA binding domain. ».
What does L7Ae RNA binding domain mean?
- In Figure 4, I first thought that there was an error in the name of the protein studied. The arrow, slightly shifted upwards by the way, indicates the position of the *Secisbp2* protein. I was confused by the fact that the protein is in most cases called *Sbp2* and rarely *Secisbp2*, a name that can be easily confounded with *secisbp2l*. Thus, it would be useful to clearly distinguish between gene and protein names and to homogenize the names in the text.
- A lower bound (< 130 kDa) is present in the *spb2* mutants compared to WT fish. Can the antibody still recognize the mutant *Sbp2* protein if the epitope is located upstream of the premature stop codon? The mutant protein, if produced and recognized by the antibody, would be what size?
- In Figure 5A, there are several problems with the form: no hyphens next to the size markers, no title on the y axis in C, no size markers and no annotation for the panel at the bottom in D. If lower panel in D shows the corresponding Coomassie staining of the upper panel, we would expect to find the *Vgt1* band in all *spb2* KO fishes as shown in B, but this is not the case. Does the loss of *spb2* not systematically result in stress?
- "Figure 6. qPCR reveals reduced selenoprotein mRNA levels in *spb2*^{-/-} but not *secisbp2l*^{-/-} larvae."
The title of Figure 6 is not correct since it shows data from RT-qPCR and mass spectrometry analysis.
- Figure 7: there is a mismatch between the designation of the graphs in Figure 7 (A-B) and its legend (A-F). Also, the blue and orange colors appear to be inverted for the "2 mM H₂O₂ - *spb2*" panel. Which mutants are studied in E (or B?) It is not very clear because there is a discrepancy between what is mentioned in the text (*secisbp2l*^{-/-}) and in the legend (*spb2*^{-/-}). Nevertheless, because *spb2*^{-/-} embryos die at 2 dpf, *secisbp2l*^{-/-} embryos are probably studied here.
- Discussion
"Thus, the N-terminal motifs that are conserved between *Sbp2* and *Secisbp2l* (Donovan and Copeland 2009) are present only in

the latter for these species." Not very clear

"Considering that ablating Secisbp2l led to a general reduction of all selenoproteins, it is more likely that selective expression in the sbp2^{-/-} background is a result of the altered mRNA levels in that strain." Not very clear

Thanks to all three reviewers for insightful comments and pointing out errors and areas that were in need of clarification. We truly appreciate the opportunity to submit an improved manuscript.

Reviewer #1 (Comments to the Authors (Required)):

Major comments:

1. It appears that the role of Secisbp2l is supplementary for SBP2, and under stress experiment, Secisbp2l mutant did not show remarkable phenotype as shown in Fig. 7A. However, the authors stated as "We propose a model where Secisbp2l is required to promote essential selenoprotein synthesis during stress." This statement includes overestimations of results.

Statement changed to "when Sbp2 activity is compromised"

2. In the introduction, the description of Secisbp2l is less. The authors should describe the previous studies related to this gene and the character of Secisbp2l similar to SBP2 here.

Such content has been added to page 3, introduction

3. The quality of Fig.1 is low and the description is insufficient. In addition, to understand this figure smoothly, the description of Secisbp2l in the introduction is helpful for readers.

We have edited the figure and legend to improve clarity

4. The quality of Fig. 2 is also low and hard to see. The authors should improve this.

We will be sure to provide high quality versions of figures for the revised submission

5. Again, the quality of Fig. 4 is low, and the arrow did not indicate a specific band precisely. In addition, the term "Secisbp2l" is described as "Secisbp2". Which is correct? The authors should check this point in the main text (for example, the title on page 16, "secisbp2-/- embryos").

Corrections have been made as suggested

6. The authors showed the high variability of selenoprotein levels in Fig. 5A. The meaning of this figure is unclear. Were these bands from sbp2-/- or includes sbp+/+ and +/-? What is the purpose of this figure? Is it correlated with other phenotypes?

We have added additional and clarifying text to this section (page 18).

7. The authors investigated the role of the Secisbp2l gene in zebrafish. Of course, this experimental system has some merit, while the experiments using cultured cells might help to understand the precise role of Secisbp2l for the synthesis of selenoproteins.

We have added text to the discussion to the effect that the use of zebrafish emanated from our inability to determine SECISBP2L function in vitro or in cells and that future work will leverage both the proximity to developmental biology that zebrafish offers and the mechanistic insight that mammalian cell culture offers to determine the mechanism of SECISBP2L action. Zebrafish appear

to be unique among vertebrate systems so far investigated in that larvae are viable without any selenoprotein synthesis.

8. The interpretation of Fig. 7 is wrong. The authors should think about the meaning of this experiment precisely.

We apologize for the confusion. The legend that indicated colors for wt versus mut was reversed.

9. As the authors mentioned, the effects of vitamin E and deferoxamine will be interesting.

10. In the discussion (page 23), the meaning of the chapter "The caveat to the idea that Secisbp2l..." was unclear. The authors should consider the obtained results precisely.

This section of the discussion has been restructured and clarified.

Reviewer #2 (Comments to the Authors (Required)):

1. It is a problem in the last sentence of the summary: "Embryos lacking Sbp2 are sensitive to oxidative stress and express the stress marker vtg1. We propose a model where Secisbp2l is required to promote essential selenoprotein synthesis during stress". This last sentence is confusing as one will understand that Secisbp2l is required to promote essential selenoprotein synthesis during oxidative stress, which is in contradiction with the results obtained. From the results part, it is more likely that it is postulated that Secisbp2l is required to promote essential selenoprotein synthesis in absence of Sbp2. Although, this point is not clear at all (see below).

We agree that this point was confusing, so we have restructured the discussion and made it clear that the function of Secisbp2l as responding to stress is purely speculative at this point.

"Additionally the muscle-specific selenoprotein SELENON was reported to be required for normal calcium flux in zebrafish embryos". This information lacks references. In addition, a comprehensive analysis of selenoprotein expression patterns in zebrafish embryos was reported (Thisse et al., 2003, Mechanism of Development-Expression Pattern, 3, 525-532). Data provided in this publication might be helpful to analyze Sbp2 and Secisbp2l deletion mutants. In addition, comparison between mined data from RNA-seq and the described selenoprotein expression patterns will be interesting.

We are grateful to the reviewer for pointing out our oversight of the prior in situ work. We have added a mention in the introduction and a section to the discussion accordingly.

The secisbp2l^{-/-} mutants were generated on a mixed TU/EK genetic background, as the sbp2^{-/-} and the double secisbp2l and sbp2^{-/-} mutants were generated on a pure EK background. The possible differences due to this genetic heterogeneity need to be mentioned and commented.

We added this point to the results section page 17.

The abbreviation dpf is not introduced.

corrected.

Mat & Meth, page 9 "Hydrogen Peroxide Treatment" - "at a density of 1000ul of embryo medium/embryo". The volume unit is not clear.

corrected.

Mat & Meth, page 10 "The digested peptides were labeled with Thermo TMTpro (Lot #: UI292951) followed by the manufacturer's protocol". This sentence sounds strange.

corrected.

Figure 1 is too small; it cannot be read properly. What are the two gene diagrams displayed in purple and green (not mentioned in the legend) In addition, the significance of the multiple lower arrows displayed at the bottom is not clear since it is mentioned in the text that a single guide RNA targeting position 2325-2344 for *secisbp2l* or position 1955-1977 for *spb2*, was used.

Corrected. We have simplified and clarified this Figure

Figure 2, a clustering of the different genes according to their expression pattern rather than displaying them by alphabetic order might help to get a clearer analyzes of the data and their categorization into different classes.

We chose to alphabetize so that paralogous gene express could be analyzed in adjacent rows with the idea that this would be the easiest way to find genes of interest.

Figure 5 needs to be properly described including top and lower panel for each case. Figure 5A, it is mentioned in the text "The significant reduction of radioactive bands in some samples corresponds to mendelian inheritance of the edited *spb2* allele (25%)". Actually on the figure, 5 lanes over the 16 analyzed showed reduced intensity; it is difficult from such sample size to deduce a statistically significant mendelian distribution.

Figure legend and text were augmented and corrected.

"In the case of *secisbp2l*^{-/-} embryos, no band-specific differences were noted when compared to wild-type embryos in the 3-5 dpf period (Figure 5C)". Actually it is Figure 5B, left panel.

Corrected.

Figure 5D, which panel is the ⁷⁵Se labeling experiment? Does the Commassie staining appear in grey scale for this experiment? In addition this CRISPR/Cas9-mediated *secisbp2l* mutagenesis in the *spb2*^{-/-} background raise questions: for the double knock-out, it appears

that most CRISPR injected animals (F0) were deleted with both secisbp2l alleles, and these embryos displayed no selenoprotein synthesis and died before 14 dpf. However, few animals survived. Were those animals depleted at a heterozygous state? Inbreeding of these heterozygous animals might help to prove and not just support the importance of both genes for full selenocysteine incorporation (not selenoprotein incorporation).

All surviving fish that were tested were wild-type. Unfortunately, 14 dpf is prior to sexual maturity in zebrafish so we will be unable to analyze the doubly deleted animals as a stable line. It is possible that we can easily create a suppressor line by making a Cys-containing version of GPX4 transgenic.

Figure 6A, the unexpected increased expression in txnrd2 and selenof mRNAs expression in the Δ secisbp2l context requires comments. Similarly, in Figure 6C, level of GPx4a detected by mass spectrometry analysis is not in agreement with the mRNA level (Figure 6A). How to interpret this result?

A deeper discussion of this topic was added to pages 24/25

Figure 7, it is a problem in panel names displayed on the figure, compared to the legend text. In addition, it is an exchange between colors used for the WT and mutant embryos.
Corrected.

Result of the last experiment combining treatment sublethal dose of peroxide and ⁷⁵Se labeling are puzzling: complete removal of selenoprotein expression in 3 and 4 dpf embryos was not dependent on the genotype (no differences between WT and Δ). It is more likely that the oxidative treatment prevent Se absorption (it is still reduced selenoprotein expression at 5 dpf) or selenocysteine insertion into proteins. How is this relevant to the hypothesis that Secisbp2l is required for "backup" selenoprotein synthesis under loss of Sbp2? To my opinion, this hypothesis should be tested in the sbp2^{-/-} background? This part is very confusing and the data obtained are overinterpreted.

We have clarified that this result did not confirm the hypothesis and does not distinguish between stage-specific loss of uptake and loss of selenoprotein synthesis. Since both the wild-type and mutant lines were affected, we did not expect to learn anything new from the sbp2^{-/-} line and elected to spare the animals in that background because we knew they were more sensitive to peroxide and would likely experience more stress.

Reviewer #3 (Comments to the Authors (Required)):

Minor remarks:

- There is no reference in the first section of the introduction about the specific selenoprotein synthesis machinery. It would be good to cite some of them.

Thanks to the reviewer for pointing out this oversight.

- "The only annotated sequence that contains a predicted sequence upstream of the conserved domains was also found in goldfish. However, the secisbp2 gene in goldfish is annotated as LOC113080603 and the corresponding mRNAs are incorrectly annotated as

secisbp2l. Interestingly, the N-terminal ~80 amino acids of the predicted goldfish Sbp2 are only found in the Clupeocephala supercohort of teleost fish, which includes the cyprinidae (zebrafish and goldfish). The N-terminal sequence that was previously found in both Sbp2 and Secisbp2l in most species (Donovan and Copeland 2009) is only present in Clupeocephala Secisbp2l but not in Sbp2."

This part is difficult to understand. It should be rewritten in a clearer manner.

Corrected (page 14).

- Nowhere is it mentioned whether the spb2 and secisbp2l genes have paralogs, as many other genes, including some selenoprotein genes, were duplicated during the third round of duplication (3R) that specifically took place in ray-finned 350 mya ago.

Corrected - added to the Results section on page 14

- In Figure 1, there are more black arrows than Crisper targets. What do the other arrows indicate?

Corrected

- Regarding figure 2, what do the numbers and the term TPM (in the text) correspond to and how was the color code established?

Added this information to text and figure legend.

- Figure 3, if it is the proteins and their different domains that are represented on the top diagrams in A and B the names on the left should not be in italics. Otherwise the diagrams represent what exactly?

Also, it would be more convenient for the comprehension to indicate the location of the premature stop codon in each case.

They are diagrams to show the relative position of known domains - changed labels to indicate protein. There are no premature stop codons in these genes as neither sbp2 nor secisbp2l are selenoproteins.

- "For Sbp2 analysis we were able to use an affinity purified commercial antibody raised against a C-terminal portion of the human protein (aa 506 - 854) which is 79% identical in the conserved L7Ae RNA binding domain. ».

What does L7ae RNA binding domain mean?

We have removed reference to this name of the domain as it does not appear elsewhere and is not relevant to the current study.

- In Figure 4, I first thought that there was an error in the name of the protein studied. The arrow, slightly shifted upwards by the way, indicates the position of the Secisbp2 protein. I was confused by the fact that the protein is in most cases called Sbp2 and rarely Secisbp2, a name that can be easily confounded with secisbp2l. Thus, it would be useful to clearly distinguish between gene and protein names and to homogenize the names in the text.

Corrected

- A lower bound (< 130 kDa) is present in the sbp2 mutants compared to WT fish. Can the antibody still recognize the mutant Sbp2 protein if the epitope is located upstream of the premature stop codon? The mutant protein, if produced and recognized by the antibody, would be what size?

It is, in fact, possible that the lower band is truncated protein, and we have added mention of this to the text on page 17.

- In Figure 5A, there are several problems with the form: no hyphens next to the size markers, no title on the y axis in C, no size markers and no annotation for the panel at the bottom in D. If lower panel in D shows the corresponding Coomassie staining of the upper panel, we would expect to find the Vtg1 band in all sbp2 KO fishes as shown in B, but this is not the case. Does the loss of sbp2 not systematically result in stress?

Corrected. Note, the Vtg1 band only strongly appears in day 3 embryos. We don't know if the band at a similar position in Figure 5D is also Vtg1...since those are all 4 dpf sbp2^{-/-} larvae, we could be looking at slight variability in ages. Note that the TCE stained image in Figure 5D is different than in the original submission because of a prior mixup in matching the stained versus phosphorimaged files.

- "Figure 6. qPCR reveals reduced selenoprotein mRNA levels in sbp2^{-/-} but not secisbp21^{-/-} larvae."

The title of Figure 6 is not correct since it shows data from RT-qPCR and mass spectrometry analysis.

Corrected

- Figure 7: there is a mismatch between the designation of the graphs in Figure 7 (A-B) and its legend (A-F). Also, the blue and orange colors appear to be inverted for the "2 mM H₂O₂ - sbp2" panel.

Which mutants are studied in E (or B?) It is not very clear because there is a discrepancy between what is mentioned in the text (secisbp21^{-/-}) and in the legend (sbp2^{-/-}).

Nevertheless, because sbp2^{-/-} embryos die at 2 dpf, secisbp21^{-/-} embryos are probably studied here.

The labeling and references were corrected.

- Discussion

"Thus, the N-terminal motifs that are conserved between Sbp2 and Secisbp21 (Donovan and Copeland 2009) are present only in the latter for these species." Not very clear

Clarified on page 23

"Considering that ablating Secisbp21 led to a general reduction of all selenoproteins, it is more likely that selective expression in the sbp2^{-/-} background is a result of the altered mRNA levels in that strain." Not very clear

This section has been expanded to add clarity.

January 11, 2022

RE: Life Science Alliance Manuscript #LSA-2021-01291-TR

Dr. Paul Copeland
Rutgers, The State University of New Jersey
Biochemistry and Molecular Biology
675 Hoes Ln
Piscataway, NJ 08854

Dear Dr. Copeland,

Thank you for submitting your revised manuscript entitled "The expression of essential selenoproteins during development requires SECIS binding protein 2-like". We would be happy to publish your paper in Life Science Alliance pending final revisions necessary to meet our formatting guidelines.

- please address the remaining reviewer 3' comments
- please upload your main manuscript text as an editable doc file
- please upload your main and supplementary figures as single files
- please add the Twitter handle of your host institute/organization as well as your own or/and one of the authors in our system
- please note that titles in the system and manuscript file must match
- please consult our manuscript preparation guidelines <https://www.life-science-alliance.org/manuscript-prep> and make sure your manuscript sections are in the correct order
- please add an Author Contributions section to your main manuscript text
- please add a conflict of interest statement to your main manuscript text
- please upload your Table in editable .doc or excel format
- we encourage you to revise the figure legend for figure 3 such that the figure panels are introduced in an alphabetical order
- please add callouts for Figures 6A and 7C-F to your main manuscript text
- please mark panel C in figure 5 and panels C, D, E, and F in figure 7
- please provide LC-MSMS data accession number in a separate Data Availability section

FIGURE CHECKS:

- there is a curve through the blots in figure 5A that looks like a splice in the blots. Please provide source data for figure 5A.

A. FINAL FILES:

- An editable version of the final text (.DOC or .DOCX) is needed for copyediting (no PDFs).
- High-resolution figure, supplementary figure and video files uploaded as individual files: See our detailed guidelines for preparing your production-ready images, <https://www.life-science-alliance.org/authors>
- Summary blurb (enter in submission system): A short text summarizing in a single sentence the study (max. 200 characters)

including spaces). This text is used in conjunction with the titles of papers, hence should be informative and complementary to the title. It should describe the context and significance of the findings for a general readership; it should be written in the present tense and refer to the work in the third person. Author names should not be mentioned.

B. MANUSCRIPT ORGANIZATION AND FORMATTING:

Sincerely,

Reviewer #1 (Comments to the Authors (Required)):

I have no concerns about this manuscript.

Reviewer #2 (Comments to the Authors (Required)):

The revisions have improved the clarity and impact of the research. I would to congratulate the authors for a excellent manuscript.

Reviewer #3 (Comments to the Authors (Required)):

I thank the authors for their responses, additions and corrections which have greatly improved the quality of the manuscript untitled "The expression of essential selenoproteins during development requires SECIS binding protein 2-like". The manuscript is now, in my opinion, acceptable for publication in Life Science Alliance, provided that a few errors that I have previously pointed out are corrected.

- Figure 5, the bottom right panel should be identified with the letter C.

- Figure 7, three problems remain:

1) There are only two letters A and B to number the two groups of panels at the top and bottom, while in the corresponding legend the panels are listed from A to F.

2) In the legend is written Wild type (orange) or mutant (blue), while the opposite is indicated in the figure.

3) in the legend "E) 3-5 dpf mutant larvae were incubated...". Which mutant is meant here? it is not indicated in the legend nor in the figure.

January 21, 2022

RE: Life Science Alliance Manuscript #LSA-2021-01291-TRR

Dr. Paul Copeland
Rutgers, The State University of New Jersey
Biochemistry and Molecular Biology
675 Hoes Ln
Piscataway, NJ 08854

Dear Dr. Copeland,

Thank you for submitting your Research Article entitled "The expression of essential selenoproteins during development requires SECIS binding protein 2-like". It is a pleasure to let you know that your manuscript is now accepted for publication in Life Science Alliance. Congratulations on this interesting work.

DISTRIBUTION OF MATERIALS:

Again, congratulations on a very nice paper. I hope you found the review process to be constructive and are pleased with how the manuscript was handled editorially. We look forward to future exciting submissions from your lab.

Sincerely,
